# Synthesis of Amino Acid–Naphthoquinones and In Vitro Studies on Cervical and Breast Cell Lines

**DOI:** 10.3390/molecules24234285

**Published:** 2019-11-25

**Authors:** Ernesto Rivera-Ávalos, Denisse de Loera, Jorge Gustavo Araujo-Huitrado, Ismailia Leilani Escalante-García, Miguel Antonio Muñoz-Sánchez, Hiram Hernández, Jesús Adrián López, Lluvia López

**Affiliations:** 1School of Chemistry, Universidad Autónoma de San Luis Potosí, Av. Manuel Nava No. 6, SLP 78210, Mexico; neto_riava@hotmail.com (E.R.-Á.); atenea.deloera@uaslp.mx (D.d.L.); 2Laboratorio de microRNAs y Cáncer, Unidad Académica de Ciencias Biológicas, Universidad Autónoma de Zacatecas, Av. Preparatoria No.301, ZAC 98068, Mexico; jalopez2210@gmail.com; 3Unidad Académica de Ciencias Químicas, Universidad Autónoma de Zacatecas, Carr. Zacatecas–Guadalajara km 6, ZAC 98160, Mexico; ileg@uaz.edu.mx (I.L.E.-G.); 27804270@uaz.edu.mx (M.A.M.-S.); hiram.hernandez.lopez@uaz.edu.mx (H.H.); 4Instituto de Investigación de Zonas Desérticas, Universidad Autónoma de San Luis Potosí, Altaír No. 200, SLP 78377, Mexico

**Keywords:** naphthoquinone, amino acids, alternative methods, microwave, ultrasound, anticancer

## Abstract

We performed an extensive analysis about the reaction conditions of the 1,4-Michael addition of amino acids to 1,4-naphthoquinone and substitution to 2,3-dichloronaphthoquinone, and a complete evaluation of stoichiometry, use of different bases, and the pH influence was performed. We were able to show that microwave-assisted synthesis is the best method for the synthesis of naphthoquinone–amino acid and chloride–naphthoquinone–amino acid derivatives with 79–91% and 78–91% yields, respectively. The cyclic voltammetry profiles showed that both series of naphthoquinone–amino acid derivatives mainly display one quasi-reversible redox reaction process. Interestingly, it was shown that naphthoquinone derivatives possess a selective antitumorigenic activity against cervix cancer cell lines and chloride–naphthoquinone–amino acid derivatives against breast cancer cell lines. Furthermore, the newly synthetized compounds with asparagine–naphthoquinones (**3e** and **4e**) inhibited ~85% of SiHa cell proliferation. These results show promising compounds for specific cervical and breast cancer treatment.

## 1. Introduction

Naphthoquinone (NQ) is a nucleus found in several natural and synthetic compounds that offers several applications, such as pigments, and many biological properties, including antibacterial, antifungal, anticancer, antimalarial, and antiviral, among others [1,2]. Regarding cytotoxic activity, several studies have reported that naphthoquinones with different substituents like paclitaxel, esters, metals, furans, carbazoles, or inclusive with carbohydrates in their structure, among others, present effects of diminishing cell proliferation [3,4,5].

These properties are principally attributed to the oxidant-reductive characteristics of the naphthoquinones, which allow the generation of dianions or semiquinone radicals. In the last decades, some publications have been dedicated to finding an explanation for the formation of these intermediates in the synthetic mechanism, and their properties that have produced different compounds with a plethora of applications of biological importance and effects that involve intra- or intermolecular interactions. One of the principal effects of these compounds is the generation of reactive oxygen species (ROS), producing cytotoxicity in different cell lines [6,7,8,9,10,11,12,13]. ROS generation with naphthoquinones represents a challenge in the design of new active compounds, principally with anticancer effect. In this context, the addition/substitution on naphthoquinone moiety by atoms or groups such as flour, oxygen, or amine can be modulated by redox properties, decreasing the toxicity levels and maintaining or potentiating the biological effect [14,15,16]. Several quinones from natural origin, like β-laphachone and menadione, among others, have been well characterized by their specific selectivity for cell lines, responding to temporal curse and dose response. Cell proliferation inhibition could be achieved by the induction of apoptosis, topoisomerase II-α inhibition, and ROS generation, among others [4]. Furthermore, it has been shown that β-lapachone inhibits epidermal growth factor (pEGFR), protein kinase B (pAKT/PKB), glycogen synthase kinase (pGsk-3β), cyclin D1, and cyclooxygenase-2 (COX-2) protein expression in a dose-dependent manner [17]. Interestingly, the naphthoquinone NSC 95,397 is used as a general potent inhibitor of cell cycle division 25C (Cdc25) [18]. Naphthoquinone regulation depends of the composition of its substituents, as well as targets and cellular components that could be inhibited and/or activated. Several naphthoquinones have been produced; however, they present non-desirable effects in cancer therapy use. Therefore, new compounds produced by alternative methods are imperative for cancer treatment.

In this regard, the synthesis and biological activities of some naphthoquinone–amino acids have been reported in several publications; nonetheless, all of these methodologies present different drawbacks, like low yields and/or long reaction times [19,20,21,22,23,24,25,26,27,28,29]. To our knowledge, the synthesis of naphthoquinone–amino acid derivatives has not been reported using microwave and ultrasound irradiation, which offer diverse advantages to conventional synthesis. In previous works, our group reported the synthesis of some Juglone and Lawsone derivatives using ultrasound and microwave irradiation under mild reaction conditions [30,31,32].

In this paper, we present the synthesis of a series of 1,4-naphthoquinone–amino acid (**3a**–**e**) and 2,3-dichloronaphhtoquinone (**4a**–**4e**) derivatives under several activation methods, such as room temperature synthesis (RTS), reflux synthesis (RS), microwave-assisted synthesis (MAS), and ultrasound-assisted synthesis (UAS), and determined the effectivity on the system. Newly synthesized compounds are promising in the cancer field. In the present work, we were able to incorporate alanine, phenylalanine, methionine, glycine, and asparagine to both naphthoquinones (**1a**,**b**) by MAS and evaluated their effect in the cervical cancer cell line, SiHa, and the breast cancer cell line, MCF-7. The incorporation of amino acids to naphthoquinones could enhance their cytotoxicity capacity, as well as their specificity.

## 2. Results

### 2.1. Chemistry

In a general method to prepare **3a**–**e** derivatives, the corresponding amino acid **2a**–**c** was added in a 1,4-type bond form to the naphthoquinone **1a**, using initially, the conditions reported in the literature (Figure 1) [19,23]. These conditions are shown as room temperature synthesis (RTS) and reflux synthesis (RS) in Table 1. Base effect was studied using potassium carbonate (K_2_CO_3_) and trimethylamine (TEA) in equimolar proportion. However, RTS and RS methodologies showed trace or no production of the desired compounds when no base was used. Only for RS and base addition were the yields increased to a range of 20–50%. A modified methodology was performed with microwave (MAS) irradiation to determine the effect on the system. Interestingly, when the system was subjected to microwave radiation, the yield of the derivatives was in the range of 28–30% without base. Furthermore, the compound yields were increased when a base was used. The results indicate that TEA and potassium hydroxide (KOH) were better bases, among others, to produce the derivative compounds (Table 1). It was determined that the optimum pH for greater product formation was 9–10. At a lower pH, the reactions became very slow, and under higher pH, the presence of secondary products was observed, as when the K_2_CO_3_ was used in an equimolar concentration (1 mmol) and not in solution.

With these established conditions, we performed the synthesis of compounds **3a**–**e** and **4a**–**e** with different bases and stoichiometric proportions between NQ and amino acids under MAS and ultrasound (UAS) as an alternative method (Table 2). It was determined that the best yields were obtained with KOH and TEA solutions. On the other hand, the yields were dependent of the proportion of each amino acid in the solution; besides, reaction times were reduced significantly to 25 min under MAS. In most cases, lower yields were obtained under UAS in comparison with MAS. Remarkably, the yield was dependent on pH, stoichiometry, and base used for each amino acid incorporation (Table 2).

In several investigations, the generation of compounds **3a**, **3b**, **3d**, **4a**, and **4d** has been reported [19,20,21,22,23,24,25,26,27,28,29], and their boiling point, infrared, and ^1^H NMR spectra have been determined. Compounds **4b**, **4c** have been reported and used as intermediates for secondary reactions or biological evaluations, but we did not find their complete characterization [33,34], while **3c**, **3e**, and **4e** have not been previously reported and, therefore, their complete characterization was carried out.

### 2.2. Electrochemical Studies by Cyclic Voltammetry

The electrochemical reduction behavior of 1,4-naphthoquinone–amino acid (**3a**–**e**) and 2,3-dichloro-1,4-naphhtoquinone derivatives (**4a**–**e**) was investigated by cyclic voltammetry (CV) in TBABF_4_ 0.1M/DMSO as supporting electrolyte at a scan rate of 100 mV·s^−1^. Figure 2A,B shows the cyclic voltammograms curves of compounds **3a**–**e** and **4a**–**e**, respectively.

In Figure 2A,B, the CV profiles of the naphthoquinone derivative compounds mainly display one quasi-reversible redox reaction process as compared to the two reduction peaks of typical quinone derivatives in aprotic media [35,36]. Interestingly, compound **3a** (Figure 2A) does not show any significant electroactive processes in the studied potential range. The irreversible reduction peak observed near to −0.8 V for **3b**–**e** and −0.9 V for **4a**–**e** in Figure 2A,B, respectively, are related to the oxidation reactions occurring by electron dislocations of the quinone derivatives at more positive potentials, >−0.7 V or >−0.5 V for both 1,4-naphthoquinone and 2,3-dichloro-1,4-naphtoquinone–amino acid derivatives, respectively. Additionally, the irreversible reduction–oxidation peaks were not observed when further cyclic voltammetry studies were performed at different scan rates in the cathodic potential region for the evaluation of the redox reactions.

The main electrochemical parameters of the redox reaction of each naphthoquinone derivative are reported in Table 3. Overall, the half-wave potentials (E_1/2_) for the redox reaction peak were near −1.24 V for **3b**–**e** and −1.12 V for **4a**–**e**, respectively (Table 3). The redox reaction is a diffusion-controlled process, since the cathodic peak current density (i_pc_) or the anodic peak current density (i_pa_) are proportional to the square root of the scan rate (i_pc_ vs. v^1/2^) [37,38].

In addition to this, the ratio of the cathodic peak current (i_pc_) to the anodic peak current (i_pa_) is close to one for the naphthoquinone–amino acid derivatives **3b**–**e** and **4a**–**e**, as described for reversible systems (Table 3). However, the potential values of cathodic peak (E_pc_) to anodic peak (E_pa_) separation (ΔE_p_) are quite large and disagree with the theoretical value reported for a one-electron reversible system, ΔE_p_ > 60 mV, as shown in Table 3 [37]. Thus, the redox reaction process is considered quasi-reversible. Furthermore, it is unclear if the reduction reaction of the quinone moiety occurs via a single-step two-electron process as compared to the two successive one-electron transfer steps producing semiquinone (Q^−^) and quinone dianion (Q^2−^), as typically observed for well-behaved quinone derivatives (Q) in nonaqueous media [35,39,40,41,42,43].

### 2.3. Effect of Naphthoquinone–Amino Acid Derivatives in Cell Proliferation of Cervical Cancer Cell Line SiHa and Breast Cancer Cell Line MCF-7

SiHa cells were treated with 0.75 mM of naphthoquinone–amino acid derivatives **3a**–**e** and **4a**–**e**; however, the concentration was extremely toxic for the cell line (data not shown). Therefore, 0.1 mM was used to register the proliferation effect on SiHa and MCF-7 cells. Compounds with glycine (**3a**) and asparagine (**3e**) substituents showed the most potent effect, inhibiting ~80% of proliferation in SiHa cells, while in MCF-7, glycine (**3a**), chloride glycine (**4a**), and chloride asparagine (**4e**) showed a proliferation inhibition near 90% (Figure 3A,B). Interestingly, the chloride substituent did not modify the effect in SiHa cells, suggesting an effect directly mediated by the amino acids glycine and asparagine (compare **3a** and **3e** versus **4a** and **4e**, Figure 3A), while in MCF-7, this substituent increased the effect of naphthoquinone–amino acid compounds (compare **3a**, **3b**, and **3c** versus **4a**, **4b**, and **4c**). In contrast, **4b** increased 10% of inhibition compared with **3b** (**4b** = 72.5% versus **3b** = 61.3%; Figure 3A). However, chloride incorporation did not always diminish proliferation, as it could be observed in **3c** versus **4c** presenting 62.6% and 54.8% of proliferation inhibition in SiHa cells, respectively (Figure 3A). Remarkably, in MCF-7 cells, chloride addition increased the proliferation inhibition effect of the compounds (Figure 3B), suggesting a specific and particular effect for breast cell line. The **3d** compound with amino acid phenylalanine showed a bigger proliferation inhibition in MCF-7 cells (57%) over SiHa cells (40%). Interestingly, for the chlorine compound **4d**, a similar effect was observed for both cell lines, presenting 69.8% of proliferation inhibition (Figure 3A,B).

## 3. Discussion

The naphthoquinone–amino acid derivatives have been previously reported; nonetheless, in this investigation, we performed a very extensive analysis about the reactivity of the equivalents of each amino acid used in the reactions. The results show a direct dependence of the pH on derivative production since, under pH = 9, reactions became very slow and above pH = 10.5, several secondary reactions appeared and reduced the yield, generating some different compounds that make difficult the purification. We optimized reactions under ultrasonic and microwave irradiation, particularly in the latter one, showing the highest efficiency obtaining the best yields 79–91% (**3a**–**e**) and 78–91% (**4a**–**e**) compared with the literature reported data, and significantly reducing the reaction times from days or hours to only a few minutes under the optimized conditions.

A full description of the electron-transfer mechanism of the naphthoquinones with amino acid substituents is beyond the scope of the present paper. However, it is assumed that the quinone dianion is possibly stabilized by an intramolecular hydrogen bond, since the dissociation of proton donor groups, i.e., carboxyl group, in DMSO is not facile, thus, cannot protonate the dianion [44,45,46,47]. In such a case, the second redox process merges with the first (Figure 2), owing to the stabilization of the dianion by a hydrogen bond, as reported elsewhere [35,46,48,49]. For instance, the E_1/2_ values computed for compounds **3b**–**e** (~1.24 V) are located at the mid-point of the half-wave potential values determined for the semiquinone (Q^−^) and the dianion (Q^2−^) redox reaction peaks of the parental 1,4-naphtoquinone compound studied at same experimental conditions, −0.9 V or −1.6 V, respectively (data not shown). Likewise, the redox reaction process peak of the naphthoquinone–amino acid and chloride derivatives (**4a**–**e**) is positioned at E_1/2_ ~ −1.12 V, as shown in Table 3, while the E_1/2_ values for the Q^−^ and Q^2–^ redox reaction steps of the 2,3-dichloro-1,4 naphthoquinone compound are −0.65 V and −1.37 V, respectively (data not show).

Particularly, naphthoquinone derivatives with glycine and chloride substituents (**4a**) present a noticeable electrochemical improvement regarding naphthoquinones with glycine (**3a**) (Figure 2). These results are associated to the electron-accepting capacity of the chloride substituent, and possibly to some conformational effects which modify the electronic configuration of the naphthoquinone–amino acid derivatives, facilitating the reduction processes [41,50]. It is then expected that naphthoquinone compounds with amino acids and chloride substituents (**4a**–**e**) present higher biological activity than derivatives (**3a**–**e**), as estimated by their E_1/2_ values, since reactive oxygen species (ROS) generation is facilitated at compounds with more positive E_1/2_ [39,40,41,42,49,51]. However, additional factors such as solvents, nature of supporting electrolyte, protonation–deprotonation equilibrium, and intra- and intermolecular hydrogen bonding also play important roles in determining the half-wave potential of the redox reaction processes, probably affecting cellular biological processes.

Recently, a proliferation inhibition effect of naphthoquinone–amino acid derivatives in different cell lines was described. In the ileocecal adenocarcinoma cell line HCT-8, it was shown an 86, 84, and 59% of proliferation inhibition with phenylalanine, alanine, and proline naphthoquinone derivatives. In the breast cancer cell line MDAMB-435, it was shown a 100, 100, and 36%, while in human multiforme cell line SF-295 86, 83, and 59% of proliferation inhibition, respectively [23]. Glycine incorporation in naphthoquinone structure is used to add methyl groups that would extend the molecules to resemble other amino acids presenting different activities [17]. Naphthoquinones with amino acids could probably interact with proteins and/or complexes, inhibiting several molecular processes like replication, transcription, and translation. Moraes et al. [23] added three complete amino acids to naphthoquinone structure, showing similar cytotoxicity to doxorubicin. The effect of glycine, alanine, and phenylalanine derivatives synthesized by Moraes et al. was similar to our results. Marastoni et al. [18] incorporated leucine, asparagine, phenylalanine, and serine into naphthoquinone with diamine alkyl spacers, inhibiting three subunits of proteasome linked to proliferation inhibition of the breast cancer cell line MDA and ovarian cancer cell line A2780 in the range of 100 μM, similar as in the present study. However, in our study, we used a number of 1 × 10^5^ cells, while they used 15 × 10^3^ cells; therefore, the molecules per cell that we used in our study was less than Marastoni et al.’s study.

The newly synthetized compounds with asparagine–naphthoquinones inhibited ~85% of SiHa cell proliferation. Inhibition grade of naphthoquinone–amino acid derivatives **3a**, **3e**, **3b**, **3c**, and **3d** ranked from 85 to 40% in SiHa cells. In contrast, MCF-7 cells presented an effect ranking from 90 to 30% of proliferation inhibition. In order from higher to lower, the compounds **4a**, **4e**, **4c**, and **4d** and **4b** showed their effect. It could be theorized that the differential effects observed may be dependent on hydropathy index and protein occurrence of the compounds. Glycine and asparagine present the negative hydropathy index −0.4 and −4.5, respectively. Nevertheless, it should be noted that asparagine’s hydropathy index is more negative that glycine’s hydropathy index, even though it is more abundant in proteins than asparagine—7.2 and 5.1, respectively. This hypothesis needs several experimental studies to be addressed. However, glycine– and asparagine–naphthoquinones, as well as dichloride compounds, show selectively antitumorigenic properties in cervix and breast cancer cell lines, respectively, positioning them as promising compounds for specific cancer treatment.

## 4. Materials and Methods

### 4.1. General

Commercially supplied 1,4-naphthoquinone, 2,3-dichloronaphthoquinone, alanine, glycine, methionine, phenylalanine, and asparagine were used for the synthesis without further purification. ^1^H and ^1^3C nucleus magnetic resonance (NMR) spectra were recorded on a Bruker Fourier 300 MHz spectrometer (^1^H at 300, ^13^C at 75 MHz, Silberstreifen, Rheinstetten, Germany) and Bruker Avance III 400 MHz spectrometer (^1^H at 400, ^13^C at 101 MHz, Silberstreifen, Rheinstetten, Germany). The spectra were acquired from solution in dimethylsulfoxide-*d*_6_ and methanol-*d*_4_ at room temperature, TMS as internal reference, the chemical shifts (δ) are expressed in part per million (ppm) and the coupling constants (*J*) in Hz. High-resolution mass spectra (HRMS) were measured with a Jeol JMS-AccuTOF through DART (Direct Analysis in Real Time, Peabody, MA, USA) and by ESI (electrospray ionization) in an Agilent 6200 Series TOF (Santa Clara, CA, USA) and 6500 Series Q-TOF LC/MS System (Santa Clara, CA, USA). Infrared spectra were recorded on a Thermo Scientific NICOLET iS10 with ATR dispositive (SMART iTR, Madison, WI, USA). Melting points were determined using a Bicote-Stuart SMO 10 apparatus (Stone, Staffordshire, UK) and were uncorrected. Microwave-assisted synthesis (MAS) was performed on a CEM Mars 6 oven with a carrousel device (Matthews, NC, USA). Ultrasound-assisted synthesis (UAS) was carried out in an Autoscience Ultrasonic cleaner-AS2060B with 60 Watts of power (Lewisville, TX, USA). TLC was performed using silica gel 60 PF_254_ containing gypsum (Merck, Darmstadt, Germany). The isolated reaction products were found to be >95% purity by NMR analysis (See Appendix A).

### 4.2. General Procedure for the Optimization of the Reaction Conditions for the Synthesis of Compounds ***3a**–**c***

A solution of the appropriate amino acid **2a**–**c** (1 mmol) in 40 mL of EtOH/H_2_O (4:1) was basified with several bases (pH = 9–10), and then a solution of 1 mmol of 1,4-naphthoquinone (**1a**) in ethanol (10 mL) was added. The mixture was activated by different methods, such as room temperature (RTS), reflux (RS), and microwave irradiation (MAS). Solvent and base effect were studied in the reaction yields, using K_2_CO_3_ (3 N) and triethylamine (TEA) (Figure 1). The progress reaction was monitored by TLC, using MeOH/CHCl_3_ (9:1) as eluent mixture. Finally, 20 mL of HCl (1 N) was added and the precipitated product was filtered and purified by flash column chromatography, starting with dichloromethane (DCM) and changing the polarity to finish with methanol.

### 4.3. Optimization of the Reaction Conditions for the Synthesis of Compounds ***3a**–**c*** Under Microwave Irradiation

In 15 mL of dioxane/H_2_O (4:1) or 30 mL of EtOH/H_2_O (4:1), 1.5 to 2 mmol of respective amino acid was added, the mixture was basified with several bases (TEA, K_2_CO_3_ aq., AcONa, or KOH aq.) to obtain a pH of 9–10, and activated under microwave irradiation with the following parameters: 110 °C, 250 W for a time of 10 min. Then, a solution of naphthoquinone (**1a**) in 5 mL or 20 mL of the respective mixture was added; the reaction solution was irradiated with the same parameters by 30 min. After the time of reaction, 20 mL of HCl (1 N) was added, and the compounds were purified with the methodology previously descripted.

### 4.4. Optimized Reaction Conditions Under Microwave Irradiation for the Synthesis of Compounds ***3a**–**e*** and ***4a**–**e***

A solution of respective amino acid (1.5–2.0 mmol) in dioxane/H_2_O (4:1, 15 mL for **3a**–**e** derivatives or 20 mL for **4a**–**e** derivatives) was basified with TEA or KOH aq. (pH 9–10) and activated under microwave irradiation (110 °C, 250 W for a time of 10 min). Then, a solution of naphthoquinone 5 mL (**1a**) or 2,3-dichloro-1,4-naphthoquinone (**1b**) 10 mL of dioxane/water mixture was added; the reaction mixture was irradiated with the same parameters by 20 min. After the time of reaction, 20 mL of HCl (1N) was added, and the compounds were purified with the methodology previously descripted. Under this methodology, compounds **3a**, **3d** and **3e**, **4b**, **4c**, and **4e** were purified as was previously described, while **3b**, **3c**, **4a**, and **4d** were obtained by filtration without further purification.

### 4.5. General Procedure for the Synthesis of Compounds ***3a**–**e*** and ***4a**–**6*** Under Ultrasonic Irradiation

For **3a**–**e** derivatives, 15 mL of dioxane/H_2_O (4:1) was used, and 20 mL for **4a**–**e** compounds. Then, 1.5–2.0 mmol of respective amino acid in solution dioxane/H_2_O was basified with TEA or KOH aq. to pH 9–10, and activated by ultrasound at 25 °C and 15 min. Then, 5 mL a solution of naphthoquinone (**1a**) or 10 mL of 2,3-dichloro-1,4-naphthoquinone (same proportion dioxane/H_2_O) was added; the reaction mixture was irradiated again for 45 min (maximum temperature of 40 °C). After the time of reaction, 20 mL of HCl (1 N) was added, and the products were purified by column chromatography with the previously described methodology.

### 4.6. Spectroscopic Characterization of Amino Acid–1,4-naphthoquinone Derivatives

2-((1,4-dioxo-1,4-dihydronaphthalen-2-yl)amino)acetic acid (**3a**). Red-brown powder. Yield: 86%. m. p.: 166–168 °C; IR (ATR): 3366, 1721, 1681, 1600, 1555, 1505, 1299, 1123 cm^−1^; ^1^H NMR (300 MHz, DMSO-*d*_6_) δ: 12.96 (s, 1H), 7.99 (dd, *J* = 7.6/0.9 Hz, 1H), 7.93 (dd, *J* = 7.6/0.9 Hz, 1H), 7.83 (td, *J* = 7.5/1.4 Hz, 1H), 7.74 (td, *J* = 7.5/1.4 Hz, 1H), 7.46 (t, *J* = 6.2 Hz, 1H, NH), 5.63 (s, 1H), 3.99 (d, *J* = 6.2 Hz, 2H) ppm. (Characterization according to literature [23].)

2-((1,4-dioxo-1,4-dihydronaphthalen-2-yl)amino) propanoic acid (**3b**). Yellow powder. Yield: 91%. m. p.: 145–147 °C; IR (ATR): 3357, 1725, 1685, 1604, 1550, 1492, 1366, 1310, 1298 cm^−1^; ^1^H NMR (300 MHz, DMSO-*d*_6_) δ: 13.08 (s, 1H), 8.00 (d, *J* = 7.5 Hz, 1H), 7.93 (d, *J* = 7.5 Hz, 1H), 7.84 (t, *J* = 7.5 Hz, 1H), 7.74 (t, *J* = 7.5 Hz, 1H), 7.27 (d, *J* = 7.6 Hz, 1H, NH), 5.66 (s, 1H), 4.21 (q, *J* = 7.3 Hz, 1H), 1.45 (d, *J* = 7.0 Hz, 3H) ppm. (Characterization according to literature [23].)

2-((1,4-dioxo-1,4-dihydronaphthalen-2-yl)amino)-4-(methylthio)butanoic acid (**3c**). Orange powder. Yield: 87%. m. p.: 146–148 °C; IR (ATR): 3339, 1716, 1676, 1600, 1550 1505 1447, 1334, 1307, 1222 cm^−1^; ^1^H NMR (DMSO-*d*_6_) δ: 13.19 (s, 1H), 8.00 (td, *J* = 7.3/1.2 Hz, 1H), 7.94 (td, *J* = 7.4/1.2 Hz, 1H), 7.84 (td, *J* = 7.5/1.4 Hz, 1H), 7.75 (td, *J* = 7.5/1.5 Hz, 1H), 7.40 (d, *J* = 8.3 Hz, 1H, NH), 5.70 (s, 1H), 4.27 (td, *J* = 8.1/5.1 Hz, 1H), 2.64–2.41 (m, 2H), 2.28–2.08 (m, 2H), 2.05 (s, 3H) ppm; ^13^C NMR (DMSO-*d*_6_) δ: 182.3, 181.7, 172.7, 148.5, 135.4, 133.2, 132.9, 130.8, 126.5, 125.8, 101.3, 54.1, 30.4, 30.1, 15.0 ppm; HRMS (DART/Q-TOF) *m*/*z*: [M + H]^+^ for C_15_H_16_NO_4_S: 306.0755; Found: 306.0801.

2-((1,4-dioxo-1,4-dihydronaphthalen-2-yl)amino)-3-phenylpropanoic acid (**3d**). Red powder. Yield: 85%. m. p.: 215–217 °C; IR (ATR): 3348, 1680, 1604, 1564, 1496, 1334, 1249, 1119 cm^−1^; ^1^H NMR (300 MHz, DMSO-*d*_6_) δ: 7.93 (d, *J* = 8.0 Hz, 1H), 7.91 (d, *J* = 8.6 Hz, 1H), 7.80 (t, *J* = 7.4 Hz, 1H), 7.69 (t, *J* = 7.4 Hz, 1H), 7.28 (d, *J* = 4.2 Hz, 1H, NH), 7.15 (m, 4H), 7.11 (s, 1H), 5.57 (s, 1H), 4.01 (q, *J* = 5.2 Hz, 1H), 3.14 (m, *J* = 7.2 Hz, 2H) ppm. (Characterization according to literature [23].)

4-amino-2-((1,4-dioxo-1,4-dihydronaphthalen-2-yl)amino)-4-oxobutanoic acid (**3e**). Dark-red powder. Yield 79%. M. p.: 147–149 °C; IR (ATR): 3438, 1734, 1682, 1591, 1528, 1379, 1276, 1231 cm^−1^; NMR ^1^H (300 MHz, DMSO-*d*_6_) δ: 7.96 (d, *J* = 7.9 Hz, 1H), 7.91 (d, *J* = 7.5 Hz, 1H), 7.81 (t, *J* = 7.0 Hz, 1H), 7.75 (s, 2H, NH), 7.69 (t, *J* = 7.2 Hz, 1H), 7.37 (d, *J* = 7.0 Hz, 1H, NH), 6.91 (s, 1H, NH), 5.67 (s, 1H), 3.93 (q, *J* = 6.3/5.8 Hz, 1H), 2.64–2.41 (m, 1H) ppm; ^13^C NMR (DMSO-*d*_6_) δ: 181.7, 181.4, 172.8, 172.7, 146.9, 134.9, 133.3, 132.2, 130.3, 125.9, 125.4, 99.8, 53.6, 38.3 ppm; HRMS (ESI/Q-TOF) *m*/*z*: [M + Na]^+^ for C_14_H_12_N_2_O_5_Na: 311.0644; Found: 311.0675.

### 4.7. Spectroscopic Characterization of Amino Acid–2,3-dichloronaphthoquinone Derivatives

2-((3-chloro-1,4-dioxo-1,4-dihydronaphthalen-2-yl)amino)acetic acid (**4a**). Orange powder. Yield: 91%. M. p.: 168–170 °C; IR (ATR): 3339, 1718, 1671 1591, 1564, 1420, 1249, 1105, 714 cm^−1^; ^1^H NMR (300 MHz, DMSO-*d*_6_) δ: 12.95 (s, 1H), 7.98 (dd, *J* = 7.4/1.4 Hz, 2H), 7.84 (td, *J* = 7.5/1.4 Hz, 1H), 7.75 (td, *J* = 7.5/1.4 Hz, 1H), 7.53 (t, *J* = 6.3 Hz, 1H, NH), 4.39 (d, *J* = 6.6 Hz, 2H) ppm. (Characterization according to literature [24].)

2-((3-chloro-1,4-dioxo-1,4-dihydronaphthalen-2-yl)amino)propanoic acid (**4b**). Orange–red powder. Yield: 95%. m. p.: 227–229 °C; IR (ATR): 3339, 1716, 1676, 1600, 1550 1505 1447, 1334, 1307, 1222 cm^−1^; ^1^H NMR (400 MHz, Methanol-*d*_4_) δ: 7.96 (dd, *J* = 7.5/3.8 Hz, 2H), 7.72 (td, *J* = 7.5/1.3 Hz, 1H), 7.63 (td, *J* = 7.6/1.2 Hz, 1H), 7.21 (d, *J* = 8.3 Hz, 1H), 4.96 (q, *J* = 6.8 Hz, 1H), 1.54 (d, *J* = 6.8 Hz, 3H) ppm; ^13^C NMR (101 MHz, Methanol-*d*_4_) δ: 186.5, 179.8, 176.7, 149.5, 134.5, 132.3, 131.4, 129.9, 126.4, 125.9, 121.1, 53.4, 20.4 ppm; HRMS (DART/Q-TOF) *m*/*z*: [M + H]^+^ for C_13_H_10_ClNO_4_: 280.0332; Found: 280.0387.

2-((3-chloro-1,4-dioxo-1,4-dihydronaphthalen-2-yl)amino)-4-(methylthio)butanoic acid (**4c**). Orange powder. Yield: 89%. m. p.: 185–188 °C; IR (ATR): 3276, 1710, 1676, 1586, 1546, 1375, 1137, 1006, 723 cm^−1^; ^1^H NMR (400 MHz, DMSO-*d*_6_) δ 8.23 (d, *J* = 7.3 Hz, 2H), 8.09 (t, *J* = 7.4 Hz, 1H), 8.01 (t, *J* = 7.5 Hz, 1H), 7.10 (d, *J* = 8.1 Hz, 1H, NH), 5.06 (q, *J* = 5.9 Hz, 1H), 2.87–2.70 (m, 2H), 2.49–2.40 (m, 1H), 2.27 (s, 3H), 1.51–1.46 (m, 1H) ppm; ^13^C NMR (101 MHz, DMSO-*d*_6_) δ 179.9, 175.4, 173.1, 143.8, 135.1, 132.7, 132.11, 129.8, 126.6, 125.9, 119.7, 56.4, 33.8, 28.9, 14.8 ppm. HRMS (DART/Q-TOF) *m*/*z*: [M + H]^+^ for C_15_H_14_ClNO_4_S: 340.0366; Found: 340.0401.

2-((3-chloro-1,4-dioxo-1,4-dihydronaphthalen-2-yl)amino)-3-phenylpropanoic acid (**4d**). Orange–red powder. Yield: 91%. m. p.: 175–177 °C; IR (ATR): 3303, 1743, 1676, 1600, 1555, 1406, 1294, 1262, 1226, 700 cm^−1^; ^1^H NMR (300 MHz, DMSO-*d*_6_) δ: 7.94 (dd, *J* = 9.0/2.9 Hz, 2H), 7.81 (td, *J* = 7.5/1.5 Hz, 1H), 7.73 (td, *J* = 7.5/1.5 Hz, 1H), 7.29–7.15 (m, 5H), 6.68 (d, *J* = 4.3 Hz, 1H), 5.32 (q, *J* = 6.2 Hz, 1H), 3.25 (dd, *J* = 5.7/2.8 Hz, 2H) ppm. (Characterization according to literature [22,29].)

4-amino-2-((3-chloro-1,4-dioxo-1,4-dihydronaphthalen-2-yl)amino)-4-oxobutanoic acid (**4e**). Red powder. Yield: 85%. m. p.: 207 °C; IR (ATR): 3442, 3280, 3195, 1671, 1626, 1595, 1559 1393, 1298, 1276, 719 cm^−1^; ^1^H NMR (300 MHz, DMSO-*d*_6_) δ: 7.94 (dd, *J* = 7.9/4.5 H, 2H), 7.81 (t, *J* = 6.1 Hz, 1H), 7.72 (t, *J* = 7.4 Hz, 1H), 7.54 (d, *J* = 3.7 Hz, 1H), 7.04 (s, 1H), 5.00–4.81 (m, 1H), 2.70–2.54 (m, 2H).; ^13^C NMR (DMSO-*d*_6_) δ: 181.7, 181.4, 172.8, 172.7, 146.9, 134.9, 133.3, 132.2, 130.3, 125.9, 125.4, 99.8, 53.6, 38.3 ppm; HRMS (ESI/Q-TOF) *m*/*z*: [M + Na]^+^ for C_14_H_11_ClN_2_O_5_Na: 345.0254; Found: 345.0287.

### 4.8. Cyclic Voltammetry

Cyclic voltammetry (CV) was performed in a conventional three-electrode electrochemical cell using a potentiostat/galvanostat (EG&G, PAR VersaStat 3). A polished glassy carbon disk electrode (GC) (0.07 cm^2^, BASi USA) was employed as working electrode. Prior to each experiment, the GC electrode was polished to mirror finishing with 0.3 mm and 0.05 mm alumina and washed with distilled water, and sonicated in ethanol for 5 min. The counter electrode was a Pt wire (Alfa-Aesar, USA), and a home-made Ag/Ag+ pseudo-reference electrode was used as a reference. The pseudo-reference electrode consisted of a silver wire in a body-glass (PINE research, USA) filled with a solution of 0.01 M silver nitrate (AgNO_3_, BASi, USA) in tetrabutylammonium tetrafluoroborate (TBABF_4_, Sigma Aldrich, USA) 0.1 M in dimethyl sulfoxide (DMSO, Sigma Aldrich, USA). The supporting electrolyte consisted of TBABF_4_ 0.1 M dissolved in DMSO. As received TBABF_4_ and DMSO were used without further purification or drying processes. Furthermore, the testing solutions were prepared by adding compounds **3a**–**e** or **4a**–**e** to the supporting electrolyte for a final concentration of 5 × 10^−3^ M. Then, cyclic voltammetry (CV) was carried out in a range from −2.0 to 0.75 V vs. Ag/Ag^+^ at a scan rate of 100 mV s^−1^ in the cathodic direction to identify any electron transfer reaction. The redox reaction processes associated to compounds **3a**–**e** or **4a**–**e** were further studied by CV at a scan rate of 10, 25, 50, 100, 200, and 300 mV·s^−1^.

### 4.9. Cell Lines

The HPV-16 positive cervical tumor line SiHa and the breast tumor cell line MCF-7 were grown in Dulbecco’s Modified Eagle’s Medium (DMEM) (Invitrogen Corporation, Carlsbad CA, United States) enriched with 5% fetal bovine serum (FBS). Medium change and passage were performed every 3 and 4 days, respectively. The SiHa cell line was kindly provided by Ph.D. Gariglio’s Lab from CINVESTAV-IPN. The MCF-7 cell line was kindly provided by Ph.D. Victor Treviño from Tecnológico de Monterrey.

### 4.10. Cell Proliferation Analysis

Cell proliferation was quantified by violet crystal dye in 1× phosphate-buffered saline (PBS) (2.7 mM KCl, 1.8 mM KH_2_PO_4_, 136 mM NaCl, 10 mM Na_2_HPO_4_ pH 7.4). The treated cells were incubated in methanol for 15 min and washed two times with water. Cells were dyed with 0.1% crystal violet and washed three times with water, and finally, violet crystal was recovered with 10% acid acetic to be analyzed in microplate reader Multiskan GO Spectrophotometer (Thermo Scientific™, Ratastic, Finland).

## 5. Conclusions

As we previously mentioned, the synthesis of naphthoquinone–amino acid derivatives has been reported; nonetheless, neither microwave- or ultrasonic-assisted synthesis has been reported. The obtained results on the analysis of the synthetic methodology showed a direct dependence of the stoichiometry of the reactants and the pH, due to the need of the amino acids for their deprotonation, and the yield was increased with the microwave system. On the other hand, some products were obtained directly by precipitation, only with the acidification of reaction mixture after a few hours of rest without further purification. Since naphthoquinones possess redox properties, cyclic voltammetry studies were done, concluding that the redox reaction is a diffusion-controlled quasi-reversible process. The biological evaluation showed a potent proliferation inhibition in cervical and breast cancer cells lines. Remarkably, naphthoquinone–amino acid derivatives, as well as dichloride compounds, showed selectively antitumorigenic properties in cervix and breast cancer cell lines, respectively, positioning them as promising compounds for specific cancer treatment.

## Figures and Tables

**Figure 1 molecules-24-04285-f001:**
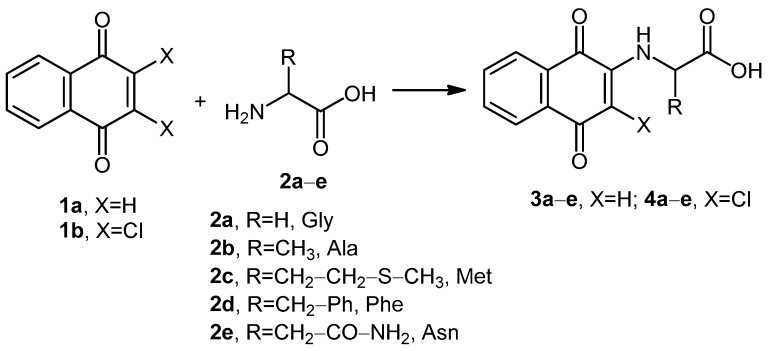
Preparation of **3a**–**e** and **4a**–**e** derivatives.

**Figure 2 molecules-24-04285-f002:**
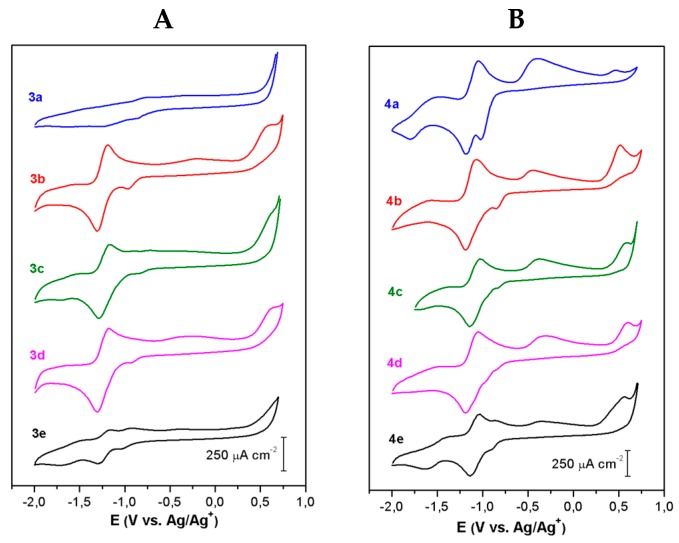
Cyclic Voltammetry curves of 5 mM naphthoquinone–amino acid derivatives (**A**) **3a**–**e** and (**B**) **4a**–**e** in TBABF_4_ 0.1 M/DMSO at 100 mV s^−1^ on a glassy carbon disk working electrode at room temperature.

**Figure 3 molecules-24-04285-f003:**
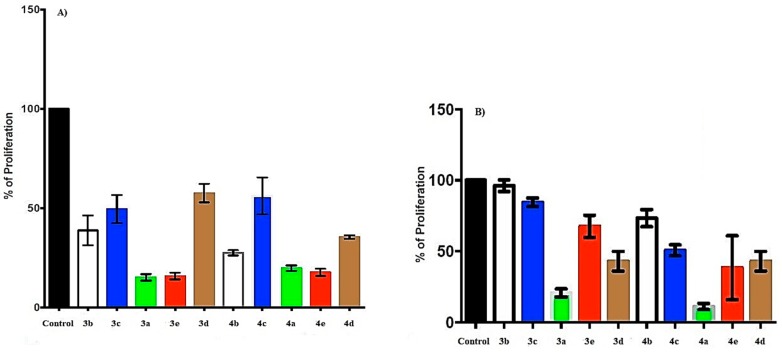
Proliferation effect of naphthoquinone–amino acid derivatives was evaluated in cancer cell lines derived from cervix and breast. (**A**) SiHa cervical cancer cells were treated with 0.1 mM of naphthoquinone–amino acid derivatives to assay proliferation rate at 72 h post-treatment. (**B**) MCF-7 breast cancer cells were treated with 0.1 mM of naphthoquinone–amino acid derivatives to assay proliferation rate at 72 h post-treatment. Cells treated with 0.001% of DMSO were used as control.

**Table 1 molecules-24-04285-t001:** Effect of reaction conditions of **3a**–**c** derivatives in their yields.

Method		Compound Yield (%)		
	Base	3a	3b	3c
RTS	None	Nr	Nr	Nr
	TEA	Tp	Tp	Tp
	K_2_CO_3_	Tp	Tp	Tp
RS	None	Tp	Tp	Tp
	TEA	32	50	30
	K_2_CO_3_	23	50	20
MAS	None	30	30	28
	TEA	74	73	71
	K_2_CO_3_	65	64	62
	AcONa	59	56	52
	KOH	74	73	72

RTS: Room temperature synthesis (25 °C for 24–48 h); RS: Reflux synthesis (90 °C for 24 h); MAS: Microwave-assisted synthesis (110 °C, 250 W, 25 min); Nr: No reaction; Tp: Trace product.

**Table 2 molecules-24-04285-t002:** Synthesis of **3a**–**e** and **4a**–**e** derivatives by MAS and UAS.

Compound	Nq–aa	MAS ^a^ (%)		UAS ^b^ (%)	
		TEA	KOH	TEA	KOH
**3a**	1:2.0	80	86	68	78
**3b**	1:1.2	85	91	75	81
**3c**	1:1.5	82	87	71	65
**3d**	1:1.5	81	85	67	78
**3e**	1:1.5	79	80	77	82
**4a**	1:2.0	91	86	86	80
**4b**	1:1.2	95	87	85	70
**4c**	1:1.5	78	89	80	68
**4d**	1:1.5	91	82	75	85
**4e**	1:1.5	78	85	75	85

^a^ MAS: 110 °C, 250 W, 25 min. ^b^ UAS: 25–40 °C, 1 h. Optimized conditions: Dioxane/water (4:1), TEA (1 mmol)/KOH (3N) 5 mL. Nq–aa: Naphthoquinone–amino acid proportion.

**Table 3 molecules-24-04285-t003:** Electrochemical parameters of naphthoquinone derivatives with amino acid substituents ^a^.

Compound	E_pa_	E_pc_	ΔE_p_ ^b^	E_1/2_ ^c^	i_pa_	i_pc_	|i_pa_/i_pc_|
	(V)	(V)	(V)	(V)	(mA cm^−2^)		
**3a**	-	-	-	-	-	-	-
**3b**	−1.18	−1.30	0.12	−1.24	0.28	−0.29	0.95
**3c**	−1.15	−1.30	0.15	−1.23	0.16	−0.17	0.93
**3d**	−1.17	−1.31	0.13	−1.24	0.28	−0.29	0.94
**3e**	−1.16	−1.32	0.16	−1.24	0.05	−0.06	0.91
**4a**	−1.04	−1.18	0.14	−1.11	0.36	−0.38	0.97
**4b**	−1.07	−1.19	0.12	−1.13	0.37	−0.41	0.90
**4c**	−1.04	−1.16	0.12	−1.10	0.27	−0.31	0.89
**4d**	−1.07	−1.20	0.13	−1.13	0.28	−0.26	1.06
**4e**	−1.05	−1.15	0.11	−1.10	0.26	−0.26	0.98

^a^ Determined by cyclic voltammetry in TBABF_4_ 0.1 M/DMSO at 100 mV/s. The potentials are given with respect to the Ag/Ag+ pseudo-reference electrode; ^b^ ΔE_p_ = E_pc_ − E_pa_; ^c^ E_1/2_ = (E_pa_ + E_pc_)/2.

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
