# Peer review of "Synthesis of Amino Acid–Naphthoquinones and In Vitro Studies on Cervical and Breast Cell Lines"

_molecules, 2019, doi:10.3390/molecules24234285_

Round 1

Reviewer 1 Report

Extensive editing of English language and style is required. There are a lot of mistakes in the text  of the manuscript and in its presentation. Provide all references according to the journal requirements. Give 1H NMR chemical shifts, not RMN. Give 13C chemical shifts to one digit after the decimal point, unless an additional digit will help distinguish overlapping peaks. The ACS Style Guide format for reporting accurate mass data is: HRMS (ESI/Q-TOF) m/z: [M + Na]+ for C13H17NO3Na 258.1101; Found 258.1074.

Author Response

Reviewer 1 Comments

Extensive editing of English language and style is required. There are a lot of mistakes in the text of the manuscript and in its presentation. Provide all references according to the journal requirements. Give 1H NMR chemical shifts, not RMN. Give 13C chemical shifts to one digit after the decimal point, unless an additional digit will help distinguish overlapping peaks. The ACS Style Guide format for reporting accurate mass data is: HRMS (ESI/Q-TOF) m/z: [M + Na]+ for C13H17NO3Na 258.1101; Found 258.1074.

Response:

The corrected manuscript was English language and style edited by an English-native speaker. The references were changed according to the journal format. The word RMN was correct to NMR. 13C NMR chemical shift were presented only with one digit after the decimal point. The format to report mass data was adjusted to ACS format.

Reviewer 2 Report

Lines 30-33 should be rewritten in order to clarify its meaning.

Line 40: "several application like pigments"

Consider changing to "several applications, such as"

If the authors are using the adjective 'several', more than one application should be included.

Line 42: Consider replacing "Taxol" for "paclitaxel"

Lines 52-54: The authors must rewrite this phrase in order to clarify its meaning.

Lines 60-62: Acronyms should be written without abbreviations in their first mention.

Lines 77-79: "The incorporation of amino acids to naphthoquinones could enhance their cytotoxicity capacity as well as their specificity." The authors must provide evidences and/or references to support this statement.

Line 74 and throughout the document: Compound numbers should be written with bold typeface. Authors should

Tables 2 and 3 should be reorganized in order to make the data presented into it more clear to the readers.

Section 2.3 and Figure 2: why no positive control was used in the same assay? This could be a very interesting place to compare your new naphtoquinones with antiproliferative compounds already in use in cancer chemotherapy.

Authors should also perform cytotoxicity (IC50) assays and a control experiment using a healthy (non-cancerous) cell line.

As stated in the Instruction for Authors: "Methods sections for submissions reporting on research with cell lines should state the origin of any cell lines." This information is not present in the manuscript.

Author Response

Reviewer 2 Comments

Lines 30-33 should be rewritten in order to clarify its meaning.

Response: lines 30-33 were rewritten.

Line 40: "several application like pigments"

Consider changing to "several applications, such as"

If the authors are using the adjective 'several', more than one application should be included.

Response: line 40 "several application like pigments” was replaced by “several application, such as”

Line 42: Consider replacing "Taxol" for "paclitaxel"

Response: line 42: “Taxol” was replaced by “paclitaxel”

Lines 52-54: The authors must rewrite this phrase in order to clarify its meaning.

Response: lines 52-54: this sentence was rewritten to improve the idea and meaning.

Lines 60-62: Acronyms should be written without abbreviations in their first mention.

Response: lines 60-62: Acronyms meaning were written in their first mention.

Lines 77-79: "The incorporation of amino acids to naphthoquinones could enhance their cytotoxicity capacity as well as their specificity." The authors must provide evidences and/or references to support this statement.

Response: lines 77-79: “The incorporation of amino acids to naphthoquinones could enhance their cytotoxicity capacity as well as their specificity”, this sentence was modified since actually it is our hypothesis considering that anilines substitution improve biological effects.

Line 74 and throughout the document: Compound numbers should be written with bold typeface. Authors should

Response: line 74: the compound numbers were written with bold typeface in all the document.

Tables 2 and 3 should be reorganized in order to make the data presented into it more clear to the readers.

Response: Tables 1, 2 and 3, were modified and the information was compiled in two tables.

Section 2.3 and Figure 2: why no positive control was used in the same assay? This could be a very interesting place to compare your new naphtoquinones with antiproliferative compounds already in use in cancer chemotherapy.

Response: We thank your comment. As you mentioned it would be interesting compare the our compounds with compounds already used, however, based on literature etoposide and paclitaxel are used at 10 mM (Li et al 2015, BMC Biochem; 16:2) and 131 mM (Aborehab et al 2019, Cancer cell Int;19:154), respectively. The new naphtoquinones competes with traditional antineoplasic compounds because we use 100 mM to set up our experiments. Besides, they use 10000 cells to do their assays in contrast to this work were we used 100000 cells. 

Authors should also perform cytotoxicity (IC50) assays and a control experiment using a healthy (non-cancerous) cell line.

Response: We agree with this suggestion, however, actually we don’t have a non-cancerous cell line.

As stated in the Instruction for Authors: "Methods sections for submissions reporting on research with cell lines should state the origin of any cell lines." This information is not present in the manuscript.

Response: This observation was already corrected.

Reviewer 3 Report

López and co-workers wish to report the functionalization of some naphthoquinones with aminoacids, electrochemical characterization of the products by cyclic voltammetry, and in vitro evaluation against breast and cervical carcinoma cell lines, just taking advantage of their interesting redox skills. Besides, the effect of various heat-sources to find optimal reaction parameters was examined. Thus, I found this manuscript within the scope of Molecules, and hence, publishable after attending some major points, as follows:

- I respectfully suggest modifying the title for ‘synthesis of aminoacid-naphthoquinones and in vitro studies on cervical and breast cell lines’

- The abstract should start in ‘We performed an extensive analysis,…’

- A deep grammar and style revision will benefit the manuscript. It could be done by an English-native speaker. In the same way, various typos were found. They must be corrected.

- The general reaction scheme would complete the tables 1 and 2. Otherwise, the reader is forced to go far away (until section 4.2).

- Indeed, tables 2 (mainly) and 3 are practically understandable/unreadable due to a large amount of footers (from a to m). Please fix them somehow.

- 13C NMR peaks must contain only one digit after dot.

- HRMS calc. must contain four digits after dot in order to gain accuracy and to measure the associated variation between calculated and found values (in ppm).

- The Supplementary Material must be completed. Various 13C NMR and Mass spectra are missing (only IR and 1H NMR are complete). Otherwise, I would not recommend publication.

Author Response

Reviewer 3 Comments:

-I respectfully suggest modifying the title for ‘synthesis of aminoacid-naphthoquinones and in vitro studies on cervical and breast cell lines’

Response: the title was modified as it was suggested.

-The abstract should be start in “We performed an extensive analysis,…’

Response: this observation was done.

A deep grammar and style revision will benefit the manuscript. It could be done by an English-native speaker. In the same way, various typos were found. They must be corrected.

Response: A grammar and style revision was done by an English-native speaker.

The general reaction scheme would complete the table 1 and 2. Otherwise, the reader is forced to go far away (until section 4.2).

Response: the general reaction scheme was moved.

-Indeed, tables 2 (mainly) and 3 are practically understandable/unreadable due to a large amount of footers (from a to m). Please fix them somehow.

Response: tables 1-3 were compiled into only two tables for better understanding.

13C NMR peaks must contain only one digit after dot.

Response: 13C NMR peaks will be changed to contain only one digit after dot.

-HRMS calc. must contain four digits after dot in order to gain accuracy and to measure the associated variation between calculated and found values (in ppm).

Response: this observation is ready corrected.

-The Supplementary Material must be completed. Various 13C NMR and Mass spectra are missing (only IR and 1H NMR are complete).

Response: for known compounds only IR and 1H NMR were done in order to structurally identify the compound.

Reviewer 4 Report

After read carefully this present manuscript, I think it is difficult for me to recommend this article for publication in this current form for publication in your journal. Thus, I need to give some comments:

By simply reading the text it is difficult to imagine the compound, the authors should add general formula on the top of each table.

1) Abstract: The first sentence should be removed or replaced in the introduction part.

2) The authors should write among their key-words "alternative methods" as a grouping expression for both Microwave and ultrasound treatments.

3) In the introduction, the authors could show the structure of the most effective compounds found in the literature. In the paragraph 2.1, which are the amine by-products ?

4) Why the authors did not use TEA for producing compounds 3a, 3b, and 3c under microwave irradiation ? (cf Table 1)

5) Under the table 2, there are too many reaction conditions depicted, it becomes difficult to understand which one is the best....

6) In my opinion, the yields obtained from ultra-sound assisted reaction are also quite good (as reported in the table 3). Maybe the authors could admit the possibility of carrying out the synthesis by this way.

7) Sometimes, in the text they have to check the size of the number( ex: TBABF4 in paragraph 2.2). always in paragraph 2., is there another method to observe the reduction of the quinone ? Sometimes comments are too theoritical, they need to be improved, especially the paragraph in conclusion from"Recently" to "15x103 cells".

8) A great part of the compounds employed in this article have been already reported elsewhere. maybe there is a lack of novelty here...

Author Response

Reviewer 4 Comments:

After read carefully this present manuscript, I think it is difficult for me to recommend this article for publication in this current form for publication in your journal. Thus, I need to give some comments:

By simply reading the text it is difficult to imagine the compound, the authors should add general formula on the top of each table.

Response: Figure 1 moved before the tables for better understanding.

1)         Abstract: The first sentence should be removed or replaced in the introduction part.

Response: the first sentence of the abstract was modified in accordance of reviewer 3.

2)         The authors should write among their key-words "alternative methods" as a grouping expression for both Microwave and ultrasound treatments.

Response: this observation was corrected.

3) In the introduction, the authors could show the structure of the most effective compounds found in the literature. In the paragraph 2.1, which are the amine by-products?

Response: we thank your comment several reviews or naphthoquinones showed biological active compounds, to avoid to be repetitive we prefer not show it.

4) Why the authors did not use TEA for producing compounds 3a, 3b, and 3c under microwave irradiation? (cf Table 1)

Response: the information is showed, to better understanding the tables 1, 2 and 3, were modified in two tables.

5) Under the table 2, there are too many reaction conditions depicted, it becomes difficult to understand which one is the best....

Response: the information of this table was rewritten.

6) In my opinion, the yields obtained from ultra-sound assisted reaction are also quite good (as reported in the table 3). Maybe the authors could admit the possibility of carrying out the synthesis by this way.

Response: although this observation is correct, we considered that MAS conditions (microwave assisted synthesis) is better compared with ultrasound because of the reaction time, in MAS it decreased considerably (25 min).

7) Sometimes, in the text they have to check the size of the number( ex: TBABF4 in paragraph 2.2). always in paragraph 2., is there another method to observe the reduction of the quinone ? Sometimes comments are too theoritical, they need to be improved, especially the paragraph in conclusion from"Recently" to "15x103 cells".

Response: these observations were corrected.

8) A great part of the compounds employed in this article have been already reported elsewhere. maybe there is a lack of novelty here...

Response: Some compounds have been already reported in literature but in low yields and with incomplete chemical characterization, and a condition methods analysis is not reported.

Round 2

Reviewer 1 Report

please find the attached file

Author Response

Thank you for your recommendations, the observations indicated in the manuscript were made, which were:

Line 44, change has by have in the sentence.

Line 68, To our knowledge,

Line 99, Nr: not reaction was changed by Nr: no reaction.

Line 207-208: Modified sentence. Recently, it has been shown a proliferation.... by: Recently, a proliferation....was described

In addition, all the numbers of the compounds were formatted in bold.

Reviewer 2 Report

The authors are strongly recommended to perform a control cytotoxicity experiment with healthy (non-cancerous) cell lines.

Author Response

Thanks for the recommendation to perform a control cytotoxicity experiment, however, actually we don’t have a non-cancerous cell line. At this time, we are preparing the anticancer assays of another series of compounds, in which we will incorporate the cytotoxicity test with the non-cancer lines. In this way, we would compensate and reference the present work.

Reviewer 3 Report

L. López and co-workers submitted a revised version of their manuscript. I found that   recommendations were attended. Now, the article is perfectly understandable.

Author Response

Thank you for you message

L. López and co-workers submitted a revised version of their manuscript. I found that   recommendations were attended. Now, the article is perfectly understandable.

Reviewer 4 Report

After read this present modified manuscript, because tha authors answered carefully to my comments, I am now able to recommend this article for publication in your journal Molecules.

Author Response

Thank you for you message:

After read this present modified manuscript, because tha authors answered carefully to my comments, I am now able to recommend this article for publication in your journal Molecules.